# Estimating Coronary Sinus Oxygen Saturation from Pulmonary Artery Oxygen Saturation

**DOI:** 10.3390/medicina60111882

**Published:** 2024-11-16

**Authors:** Alexander Gall, Hosamadin S. Assadi, Rui Li, Zia Mehmood, Bahman Kasmai, Gareth Matthews, Pankaj Garg

**Affiliations:** 1Department of Cardiology, Addenbrooke’s Hospital, Cambridge University Hospitals NHS Foundation Trust, Cambridge CB2 0QQ, UK; 2Department of Cardiovascular and Metabolic Health, Norwich Medical School, University of East Anglia, Norwich NR4 7TJ, UK; 3Department of Cardiology, Norfolk and Norwich University Hospitals NHS Foundation Trust, Norwich NR4 7UY, UK

**Keywords:** coronary sinus, oxygen saturation, coronary sinus flow, cardiac MRI

## Abstract

*Background and Objectives*: Coronary sinus oxygen saturation is a useful indicator of health and disease states. However, it is not routinely used in clinical practice. Cardiovascular magnetic resonance imaging (CMR) oximetry can accurately estimate oxygen saturation in the pulmonary artery. This research aimed to provide a method for calculating coronary sinus oxygen saturation (ScsO_2_) from pulmonary artery oxygen saturation (SpaO_2_) that could be applied to CMR. *Materials and Methods*: A systematic literature review was conducted to identify prior work that included invasive measures of ScsO_2_ and either SpaO_2_ or right ventricular oxygen saturation. This revealed one study with appropriate data (ScsO_2_ and SpaO_2_ measurements, *n* = 18). We then carried out agreement and correlation analyses. *Results*: Regression analysis demonstrated a statistically significant, positive relationship between ScsO_2_ and SpaO_2_, giving a regression equation of ScsO_2_ = −31.198 + 1.062 × SpaO_2_ (*r* = 0.76, *p* < 0.001). A multivariable regression analysis of all reported variables, excluding SpaO_2_, independently identified superior vena cava oxygen saturation (SsvcO_2_) and arterial oxygen saturation (SaO_2_) as predictors of ScsO_2_ (*r* = 0.78, *p* < 0.001), deriving the equation ScsO_2_ = −452.8345 + 4.3579 × SaO_2_ + 0.8537 × SsvcO_2_. *Conclusions*: In this study, we demonstrated a correlation between coronary sinus oxygen saturation and pulmonary artery oxygen saturation, allowing the estimation of ScsO_2_ from SpaO_2_. This association enables the estimation of ScsO_2_ from purely CMR-derived data. We have also described a second model using arterial and superior vena cava saturation measurements, providing an alternative method. Future validation in larger, independent cohorts is needed.

## 1. Introduction

The coronary sinus is the most prominent cardiac vein, responsible for returning approximately 90% of the coronary blood supply to the right atrium, with the remaining 10% drained by the Thebesian veins [1]. Coronary sinus oxygen saturation (ScsO_2_) is a critical parameter in the comprehensive assessment of myocardial oxygenation and metabolic status, offering clinicians valuable insights into cardiac function across various health and disease states [2,3]. As a direct indicator of the oxygen saturation of blood returning from the myocardium through the coronary sinus, ScsO_2_ provides a unique and essential perspective on the balance between myocardial oxygen supply and demand [1,4]. This balance is crucial for determining the adequacy of myocardial perfusion, giving ScsO_2_ measurement the potential to be an important tool in diagnosing and managing a wide range of cardiac conditions, including during and following cardiac catheter intervention and cardiac surgery [2,5,6,7,8].

ScsO_2_ is of particular importance because it reflects myocardial oxygen extraction. Low ScsO_2_ values typically suggest increased oxygen extraction by the myocardium or a decreased oxygen supply, both of which can indicate underlying cardiac issues [4]. By serving as an early warning system for potential myocardial ischaemia, ScsO_2_ measurement has the potential to enable clinicians to identify and address cardiac diseases at an earlier stage, preventing their progression into more severe conditions. Such early intervention is critical for improving patient outcomes, as timely treatment can prevent further damage to the heart muscle and reduce the likelihood of complications [4,9].

The clinical utility of ScsO_2_ extends beyond diagnostic applications to intraoperative monitoring, where it has proven especially beneficial [10]. During cardiac surgery, real-time monitoring of ScsO_2_ allows for the detection of unrecognised myocardial ischaemia [10]. During an coronary artery bypass graft (CABG), ScsO_2_ monitoring helps assess the success of revascularisation efforts, ensuring that the myocardium receives adequate perfusion post-surgery [7,10]. By providing this level of detailed, real-time information, ScsO_2_ monitoring enhances patient safety and can improve procedural outcomes [10,11]. Furthermore, in percutaneous coronary intervention (PCI), continuous ScsO_2_ monitoring can alert clinicians to ischaemic events as they occur, allowing for prompt adjustments that can prevent significant myocardial damage or procedure-related myocardial infarction [6,8,12].

Beyond its role in intraoperative monitoring, ScsO_2_ holds substantial prognostic value in managing patients with known or suspected coronary artery disease (CAD) [11,13]. Studies have shown that ScsO_2_, in combination with measures of resting coronary sinus blood flow (CSBF) and coronary flow reserve (CFR), measured via cardiovascular magnetic resonance imaging (CMR), can predict major adverse cardiac events (MACE) [11,13]. In a study of 693 patients with either CAD or suspected CAD, Kato et al. not only demonstrated the feasibility of measuring CSBF by two-dimensional phase contrast velocity encoded imaging but also showed that a higher resting CSBF, in particular, is associated with poorer outcomes, irrespective of the presence of myocardial ischaemia or not [13]. The plausible explanation for their finding was that the CSBF directly contributes to total myocardial oxygen consumption. In effect, the study was sub-phenotyping patients with higher total myocardial oxygen consumption. Importantly, the most striking finding of their study was that the prognostic role of CSBF was independent of scar assessment by late gadolinium enhancement and myocardial ischaemia assessment by first-pass perfusion assessment [11,13].

Integrating ScsO_2_ monitoring into routine clinical practice holds significant potential to enhance our understanding of myocardial physiology and pathophysiology. As more data on ScsO_2_ is collected and analysed, new patterns and trends will likely emerge, providing deeper insights into the complex relationship between myocardial oxygen supply and demand. This understanding will likely pave the way for discoveries of better sub-phenotyping patients with heart failure and cardiac metabolic disease, possibly improving early detection, treatment monitoring, and prognostication.

Despite the evident clinical importance of ScsO_2_, traditional methods for measuring this parameter are invasive, requiring catheterisation to sample blood from the coronary sinus directly before processing samples using a blood gas analyser [14]. Although these techniques provide highly accurate, real-time data, they carry significant risks, such as bleeding and infection [14]. Moreover, the need for specialised equipment and trained personnel renders these methods resource-intensive and less accessible, mainly when repeated measurements are necessary. While ScsO_2_ measurement may be feasible with minimal additional intervention in patients already undergoing a procedure, its utility as a diagnostic marker is limited in patients who do not require catheterisation for other reasons. This limitation underscores the need for developing non-invasive methods for measuring ScsO_2_, which could greatly enhance the diagnosis and management of a broader patient population. Pulse oximetry provides a non-invasive measure of oxygen saturation, utilising the absorptive properties of oxyhaemoglobin to red and infrared light. However, this only allows for a representation of the arterial SO_2_ rather than more targeted assessments of oxygen saturation [15]. In recent years, there has been growing interest in non-invasive measurement techniques that can provide the same valuable information as invasive methods but with fewer risks and greater patient comfort. For instance, positron emission tomography (PET) imaging has been used to non-invasively assess myocardial blood flow and oxygenation [16]. However, PET imaging requires ionising radiation, which carries its own set of risks, particularly with repeated exposure [16]. Consequently, the search for safer, non-invasive alternatives continues.

CMR has emerged as a promising non-invasive alternative that does not require ionising radiation. T2-based CMR oximetry, in particular, can accurately estimate oxygen saturation at various locations, such as the pulmonary artery and the ascending aorta [16]. Varghese et al. (2020) compared CMR oximetry with invasive measures of oxygen saturation on right heart catheterization in 32 patients, alongside 5 healthy controls undergoing CMR examination only, focusing on the pulmonary artery and SVC. The authors concluded that CMR oximetry was repeatable and reproducible, with good agreement with invasive measures [17]. The coronary sinus, however, is too small to accurately measure using this technique on a routine CMR acquisition [14,18]. This limitation of current CMR technology highlights the need for a different strategy to make non-invasive ScsO_2_ measurement more feasible and reliable. Developing a reliable non-invasive method for estimating ScsO_2_ would represent a major advancement in the field of cardiac monitoring, offering a safer and more accessible approach to assessing this parameter.

Given the challenges associated with both invasive and non-invasive methods, our study aims to bridge this gap by developing a model to estimate coronary sinus blood oxygen saturation from measured pulmonary artery oxygen saturation. This model will be based on previously published data involving invasive sampling at both locations, with the aim of then applying it to non-invasive measures of pulmonary artery oxygen saturation allowing for the non-invasive estimation of coronary sinus oxygen saturation.

## 2. Materials and Methods

### 2.1. Study Design

This study was conducted as a retrospective observational analysis, utilising existing data derived from previously published open-access research. The datasets analysed were sourced from studies that had been made publicly available, ensuring transparency and accessibility. These original datasets were collected in strict adherence to ethical standards and regulatory guidelines, including informed consent where applicable, and were subsequently published in peer-reviewed journals. In this analysis, no new data were collected, and all data processing and analyses were performed on this pre-existing, publicly accessible information.

### 2.2. Search Strategy

A comprehensive and systematic search strategy was employed on 10 July 2024 to identify relevant studies for this analysis. PubMed, Scopus, and Web of Science databases were extensively searched using a strategic combination of Medical Subject Headings (MeSH) terms and free-text keywords. The primary search terms included “oxygen saturation”, “pulmonary artery”, “coronary sinus”, “coronary venous”, and “human”. To refine the search strategy and ensure precision, Boolean operators (AND, OR) were employed. Additionally, filters were applied to limit the search results to peer-reviewed articles published in the English language.

Beyond database searches, the reference lists of all identified articles were meticulously hand-searched to uncover additional studies that met our predefined inclusion criteria. This approach ensured a comprehensive collection of relevant studies.

Potentially relevant studies were initially screened by title and abstract for appropriateness. The studies were published between 2004 and 2024. Studies deemed relevant based on the initial screening underwent a thorough full-text review. Of all the studies screened, only one provided published data detailing oxygen saturation levels in the coronary sinus and the pulmonary artery.

### 2.3. Study Data

Data extraction focused on specific variables from the identified study by Bouchacort et al. [19]. We concentrated on data presented in Table 2 of their paper, specifically targeting oxygen saturation levels in the pulmonary artery and the coronary sinus. Additionally, we extracted data on oxygen saturation levels in the superior vena cava (SsvcO_2_), inferior vena cava (SivcO_2_), and the systemic arterial system (SaO_2_).

To ensure accuracy and compliance, we communicated with the original study’s publisher. They confirmed that no further permissions were required for our detailed examination of the original data. This process ensured that our use of the data was both ethical and precise.

This study adhered to the Strengthening the Reporting of Observational Studies in Epidemiology (STROBE) guidelines [20], ensuring robust reporting standards for observational studies. Additionally, the transparent reporting of a multivariable prediction model for Individual Prognosis Or Diagnosis (TRIPOD) guidelines were followed [21], ensuring thorough and transparent reporting for studies developing prediction models.

### 2.4. Statistical Analysis

All statistical analyses were performed using MedCalc (MedCalc^®^ Statistical Software version 22.009 (2023), Ostend, Belgium: MedCalc Software Ltd.). Normal distribution was tested using the Shapiro–Wilk test. Continuous variables were summarised using the mean value ± standard deviation (SD). Categorical data were expressed as frequencies and percentages (%). The analysis centred on selecting relevant variables for a backward multivariate linear regression. Variables with significant correlations were included in the regression model development. The strength and direction of the linear relationships between variables were evaluated using the Pearson product-moment correlation coefficient (r). Additionally, bias in the analysis was evaluated using Bland–Altman plots, which helped in assessing agreement between two quantitative measurements.

Statistical significance was rigorously defined as a *p*-value of less than 0.05. This threshold ensured that the results obtained were statistically meaningful and reliable.

## 3. Results

The study identified by Bouchacort et al. sought to examine the oxygen saturation gradient between the superior vena cava and the pulmonary artery. The authors hypothesised that deoxygenated blood from the myocardium entering via the coronary sinus would significantly contribute to this gradient [19]. Their study conducted simultaneous blood sampling at four sites: the coronary sinus, pulmonary artery, superior vena cava, and inferior vena cava. This procedure was performed in a cohort of 18 patients under general anaesthesia for cardiac surgery. The mean ± standard deviation (SD) oxygen saturation levels at these sites were as follows: the ScsO_2_ (coronary sinus) was 45.5 ± 17%, the SpaO_2_ (pulmonary artery) was 72.2 ± 12%, the SsvcO_2_ (superior vena cava) was 76.6 ± 12.6%, the SivcO_2_ (inferior vena cava) was 73.2 ± 15.9%, and the SaO_2_ (systemic arterial system) was 99.3 ± 1.4% (Figure 1). Additionally, this patient cohort’s mean ± SD cardiac index was 1.9 ± 0.5 mL/min/m^2^. The comprehensive results of these measurements are presented in Table 1.

### 3.1. Bland–Altman Analysis

Figure 1 illustrates the Bland–Altman plot comparing the oxygen saturation levels in ScsO_2_ and SpaO_2_. This analysis included a total of 18 paired measurements for each variable, as detailed in Table 1. The Bland–Altman plot revealed a mean difference of 26.7% between the ScsO_2_ and SpaO_2_, with a 95% confidence interval (CI) ranging from 21.2% to 32.2%. The *p*-value was <0.0001, indicating a statistically significant difference. The plot also showed 95% limits of agreement from 4.9% to 48.5%, suggesting a consistently higher SpaO_2_ than ScsO_2_. This consistent and significant positive arithmetic mean difference underscores the gradient between these two sites.

### 3.2. Regression Analysis

A backward multivariable regression analysis was conducted to predict the oxygen saturation in the ScsO_2_, incorporating the oxygen saturation values reported at all sites by Bouchacort et al.: the SpaO_2_, SsvcO_2_, SivcO_2_, and SaO_2_ [19]. This analysis identified SpaO_2_ as the only significant independent variable. A univariate regression analysis was performed to explore this relationship further, which included all 18 observations from the study (Table 1) [19]. The resulting regression equation is as follows (Figure 2):ScsO_2_ = −31.198 + 1.062 × SpaO_2_

The slope of the regression line was 1.0624 with a standard error of 0.2288 and a 95% confidence interval ranging from 0.5773 to 1.5475. The intercept was −31.1978 with a standard error of 16.739 and a 95% confidence interval from −66.6829 to 4.2872. The *p*-value for the slope was <0.001, indicating a statistically significant relationship between the two variables (Figure 3).

The correlation coefficient (*r*) for this model was 0.76, demonstrating a strong positive linear relationship between SpaO_2_ and ScsO_2_ (*p* < 0.001).

The Analysis of Variance (ANOVA) for the regression model showed a significant F-ratio of 20.1082 (*p* < 0.001), further supporting the significance of the model. Additionally, the residuals of the model were tested for normality using the Shapiro–Wilk test, which yielded a W-value of 0.9332 and a *p*-value of 0.221, indicating that the residuals followed a normal distribution.

In an alternative regression model, SpaO_2_ was excluded from the analysis to assess the predictive power of other variables. Backward multivariable regression was conducted to predict ScsO_2_, starting with the remaining variables. The initial model included SaO_2_ (β = 0.36, *p* = 0.047), SsvcO_2_ (β = 0.39, *p* = 0.236), and SivcO_2_ (β = 0.28, *p* = 0.384). Due to its lack of statistical significance, SivcO_2_ was excluded from the final model. The refined model identified SaO_2_ and SsvcO_2_ as significant independent predictors of ScsO_2_. Specifically, SaO_2_ had a coefficient of β = 0.35 (*p* = 0.047), and SsvcO_2_ had a coefficient of β = 0.63 (*p* = 0.002). Table 2 summarises both models, with this model being denoted as “Model 2”.

The resulting regression equation for the final model was
ScsO_2_ = −452.8345 + 4.3579 × SaO_2_ + 0.8537 × SsvcO_2_

This model accounted for a substantial portion of the variance in ScsO_2_, with an R^2^ value of 0.612 and an adjusted R^2^ of 0.56. The standard error of the estimate was 11.278, reflecting the typical deviation of the observed values from the predicted values.

The overall significance of the model was further confirmed by ANOVA, which showed an F-ratio of 11.839 (*p* < 0.001), indicating that the model significantly predicts ScsO_2_ better than a model with no predictors.

In summary, the final regression model demonstrated that both SaO_2_ and SsvcO_2_ are significant predictors of ScsO_2_, providing a robust framework for understanding the factors influencing ScsO_2_. Further diagnostic tests confirmed the model’s assumptions, ensuring the reliability and validity of the results. This comprehensive analysis underscores the importance of considering multiple variables when assessing oxygen saturation levels in different parts of the cardiovascular system.

## 4. Discussion

We demonstrated a significant positive correlation between coronary sinus oxygen saturation and pulmonary artery oxygen saturation. Since SpaO_2_ is easier to measure using routine T2 CMR oximetry, it can be used to non-invasively estimate ScsO_2_, reducing the procedural risks of invasive measurements. We then go on to describe a method for estimating ScsO_2_ from arterial and superior vena caval oxygen saturation (SaO_2_ and SsvcO_2_, respectively). This may prove useful in clinical practice if SpaO_2_ measures are unavailable, with SaO_2_ available from routine oximetry and SsvcO_2_ measurable using CMR oximetry.

Using the Fick principle, myocardial oxygen consumption (MVO_2_) can be calculated by combining myocardial blood flow with the arteriovenous oxygen concentration difference. Our method allows for a non-invasive estimation of ScsO_2_, which can be incorporated into this calculation to derive MVO_2_ using CMR. However, avoiding significant confounding conditions, such as cardiac shunts or peripheral vascular disease, is crucial, as it could affect measurement accuracy.

For patients in the postoperative period following cardiac surgery, assessing ScsO_2_ and MVO_2_ can be valuable in identifying ongoing ischaemia or inadequate reperfusion [2,10]. Providing a non-invasive means of assessment, using routine CMR acquisitions, provides a potential opportunity to assess the success of a procedure and identify those who may require re-intervention or closer follow-up and monitoring.

The non-invasive estimation of coronary sinus oxygen saturation could significantly impact the management of patients with heart failure, a condition which involves impaired myocardial perfusion and increased myocardial oxygen demand, which are critical factors influencing patient prognosis. By providing a non-invasive means to estimate ScsO_2_, clinicians could more readily assess the balance between myocardial oxygen supply and demand, thereby evaluating the effectiveness of medical therapies in real time and quantitatively. This approach could also facilitate the early detection of deteriorating cardiac function, allowing for timely adjustments in treatment and potentially improving outcomes by preventing acute decompensations and reducing hospitalisations.

In patients with chronic coronary syndromes, non-invasive ScsO_2_ estimation offers a valuable tool for ongoing monitoring of disease progression and therapeutic efficacy. Chronic coronary syndromes often involve stable but reduced myocardial perfusion, where small changes in oxygen supply or demand can indicate worsening ischaemia. Assessing ScsO_2_ could help identify subtle declines in myocardial oxygenation, prompting earlier medical intervention before clinical symptoms worsen. Furthermore, this method could aid in evaluating the effectiveness of anti-ischaemic therapies by providing objective data on their impact on myocardial oxygenation. This could enable more personalised management strategies by quantifying physiological responses to treatment.

Incorporating non-invasive ScsO_2_ estimation into stress testing protocols, such as dobutamine stress CMR, could provide additional insights into myocardial oxygen demand and reserve. Traditional stress testing methods primarily focus on wall motion abnormalities or perfusion deficits when investigating myocardial ischaemia. By integrating ScsO_2_ measurements, clinicians could gain a deeper understanding of how the myocardium responds to increased workloads, identifying ischaemic thresholds more precisely. This could improve the sensitivity and specificity of stress tests, particularly in detecting subclinical ischaemia, and might lead to better risk stratification and treatment planning in patients with suspected coronary artery disease, with the potential for earlier medical therapy.

Furthermore, derivation of ScsO_2_ and assessment of MVO_2_ in acutely unwell patients in an intensive care setting has the potential to help guide treatment decisions as part of an overall approach to physiological monitoring, especially in patients undergoing pulmonary arterial catheterisation. A rising myocardial oxygen demand could indicate a deteriorating patient and therefore the need to titrate or escalate treatment.

In summary, the assessment of coronary sinus oxygen saturation provides a useful parameter that has clinical applications across a variety of cardiac and non-cardiac diseases to help guide treatment, monitor responses to treatment and assess the success of interventions.

### Limitations

This study has a small sample size and is of a retrospective analysis. This makes the results hypothesis-generating rather than conclusive. The primary limitation of this study is the lack of an independent dataset to validate our equation. As such, this work must be interpreted in this context until further validation can be achieved. Despite an extensive literature review, no other studies provided simultaneous measurements of coronary sinus and pulmonary artery oxygen saturations. Our equation assumes the constant saturation of systemic venous blood and that coronary sinus blood mixing is the primary determinant of pulmonary arterial saturations. Therefore, the equation may not hold under varying systemic oxygen consumption conditions, such as in sepsis or during exercise. Future research should validate this equation in larger, independent patient cohorts. It would be of equal importance to test this equation in disease states prior to application to these patient cohorts. Once validated, this method could be applied to CMR datasets to estimate MVO_2_, aiming to correlate these estimates with various health and disease states.

## 5. Conclusions

In conclusion, our study shows that coronary sinus oxygen saturation can be estimated from pulmonary artery oxygen saturation measurements. We further demonstrate means of estimating coronary sinus oxygen saturation from arterial and superior vena caval oxygen saturation measurements.

## Figures and Tables

**Figure 1 medicina-60-01882-f001:**
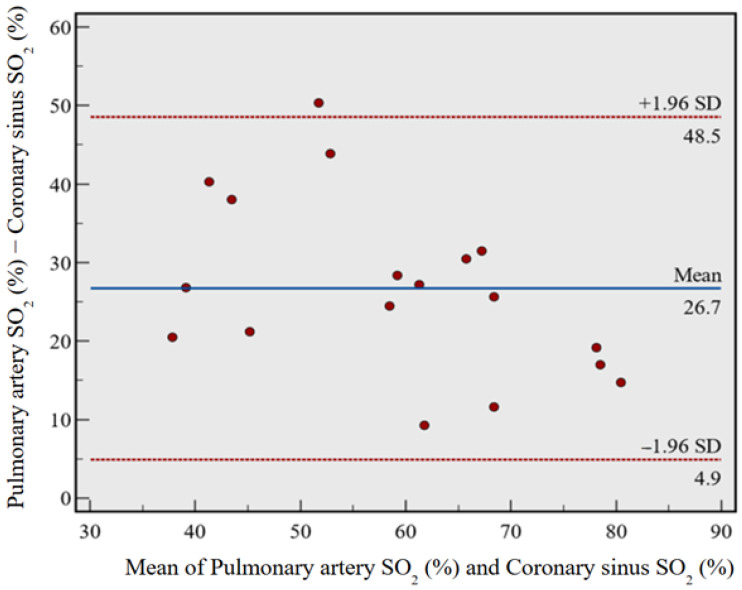
Bland–Altman plot comparing pulmonary artery oxygen saturation and coronary sinus oxygen saturation. There was a mean difference of 26.7% (*p* < 0.0001), suggesting a consistent SO_2_ gradient between SpaO_2_ and ScsO_2_. SO_2_ = oxygen saturation.

**Figure 2 medicina-60-01882-f002:**
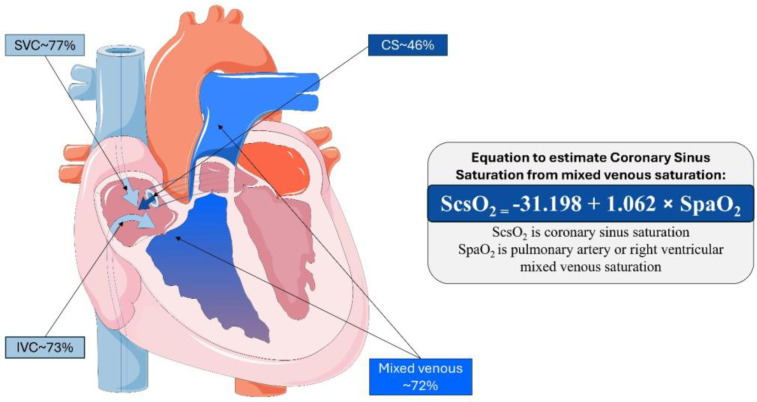
Central illustration demonstrating the contribution to right ventricular/pulmonary artery saturation from the caval system and coronary sinus, respectively. Mean oxygen saturation is derived from Table 1. The equation is derived from multivariable regression analysis to estimate ScsO_2_ from measured SpaO_2_. SVC = superior vena cava. IVC = inferior vena cava. CS = coronary sinus.

**Figure 3 medicina-60-01882-f003:**
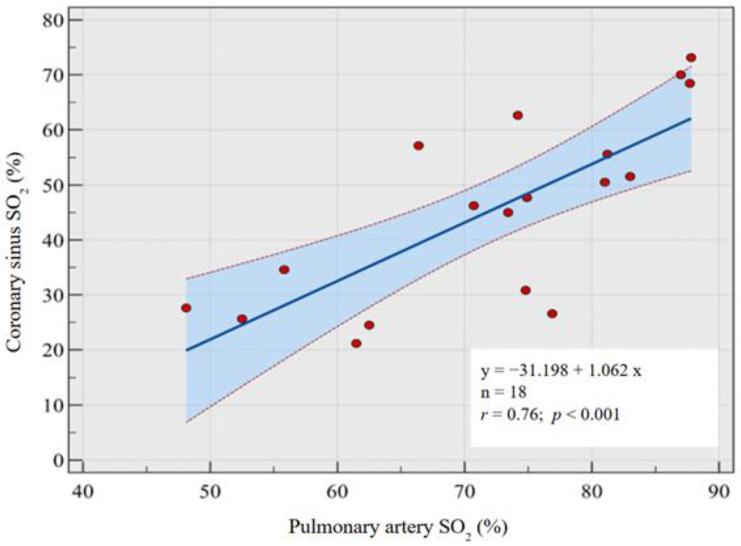
Correlation coefficient between coronary sinus oxygen saturation and pulmonary artery oxygen saturation, allowing the derivation of a model to estimate coronary sinus oxygen saturation from measured pulmonary artery oxygen saturation. SO_2_ = oxygen saturation.

**Table 1 medicina-60-01882-t001:** Results of saturation measurements from simultaneous blood sampling at the coronary sinus and pulmonary artery, adapted from the study by Bouchacort et al. [19].

*Patient*	*ScsO_2_ (%)*	*SpaO_2_ (%)*	*SsvcO_2_ (%)*	*SivcO_2_ (%)*	*SaO_2_ (%)*
*1*	26.6	76.9	80.3	82.2	99.7
*2*	21.2	61.5	72.1	72.2	94.0
*3*	68.5	87.7	96.1	88.9	99.6
*4*	51.5	83.0	85.1	89.4	99.7
*5*	27.6	48.1	53.1	48.7	99.4
*6*	30.9	74.8	66.1	79.2	99.4
*7*	45.0	73.4	85.7	70.9	100.0
*8*	34.6	55.8	55.0	49.2	99.9
*9*	24.5	62.5	65.9	55.8	99.0
*10*	46.2	70.7	74.7	58.7	99.8
*11*	57.1	66.4	68.4	72.6	99.8
*12*	25.7	52.5	64.1	44.8	98.9
*13*	73.1	87.8	89.8	91.1	100.0
*14*	62.6	74.2	80.1	77.0	99.6
*15*	50.5	81.0	86.9	85.3	99.9
*16*	55.6	81.2	94.0	92.9	99.9
*17*	47.7	74.9	77.4	69.4	99.7
*18*	70.0	87.0	84.2	89.7	99.9

ScsO_2_ = coronary sinus oxygen saturation. SpaO_2_ = pulmonary artery oxygen saturation. SsvcO_2_ = superior vena cava oxygen saturation. SivcO_2_ = inferior vena cava oxygen saturation. SaO_2_ = arterial oxygen saturation.

**Table 2 medicina-60-01882-t002:** Summary of multivariable linear regression model fitting. The 95% confidence intervals for each variable included are shown in brackets.

	ScsO_2_ (%) Model 1		ScsO_2_ (%) Model 2	
Model Constant	−31.198	(−66.683 to 4.287)	−452.835	(−878.783 to −26.886)
Variable 1 Coefficient	−1.062	(0.577 to 1.548)	4.358	(0.011 to 8.704)
Variable 1	SpaO_2_ (%)		SaO_2_ (%)	
Variable 2 Coefficient			0.854	(0.381 to 1.327)
Variable 2			SsvcO_2_ (%)	
Excluded Variables	SivcO_2_ (%) SsvcO_2_ (%) SaO_2_ (%)		SivcO_2_ (%)	
Adjusted R^2^	0.547		0.56	
F-statistic	21.551		11.839	
Significance	<0.001		<0.001	

ScsO_2_ = coronary sinus oxygen saturations. SpaO_2_ = pulmonary artery oxygen saturations. SaO_2_ = systemic arterial oxygen saturations. SsvcO_2_ = superior vena cava oxygen saturations. SivcO_2_ = inferior vena cava oxygen saturations.

## Data Availability

The original data presented in this study are available in Minerva Anestesiologica at https://www.minervamedica.it/en/journals/minerva-anestesiologica/article.php?cod=R02Y2011N06A0579 (accessed on 11 November 2024), Table 2. These data were derived from the resource available in the public domain at the URL provided.

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
