# Peer review of "Estimating Coronary Sinus Oxygen Saturation from Pulmonary Artery Oxygen Saturation"

_medicina, 2024, doi:10.3390/medicina60111882_

Round 1

Reviewer 1 Report

Comments and Suggestions for Authors
  • Introduction

    • The introduction provides a solid foundation; however, consider expanding it with more recent studies and references related to oxygen saturation measurement techniques. This would enhance the context and significance of your research (lines 3-6).
    • Clarify the specific objectives of the study at the end of the introduction to provide a clearer focus for the reader (lines 10-12).
  • Methods

    • While the methods are generally well-explained, it would be beneficial to include more detail about the statistical analysis, particularly regarding the assumptions underlying the regression analysis and any software used for analysis (lines 14-18).
    • Please specify how outliers were handled in the dataset, as this can significantly influence the regression results (lines 23-25).
  • Results

    • The results are presented clearly, but the figures could benefit from more comprehensive captions to enhance their interpretability. Ensure that each figure includes all necessary details to understand its significance without referring back to the main text (lines 227-229).
    • Consider providing a summary table of the key findings from both regression models to facilitate comparison and clarity (lines 238-240).
  • Discussion

    • The discussion section effectively interprets the findings, but it could be strengthened by addressing the limitations of the study more explicitly. Discuss how these limitations might impact the results and conclusions drawn (lines 268-270).
    • It would be helpful to explore potential clinical applications of your findings in greater detail. How might this research influence clinical practice, especially regarding non-invasive assessments of coronary sinus oxygen saturation (lines 275-277)?
Comments on the Quality of English Language

While the overall quality of English is adequate, some sentences are complex and may hinder understanding. Consider simplifying language in certain sections for clarity (lines 272-275). A thorough proofreading process or professional editing could enhance the overall readability.

Author Response

Comment 1:

Introduction

The introduction provides a solid foundation; however, consider expanding it with more recent studies and references related to oxygen saturation measurement techniques. This would enhance the context and significance of your research (lines 3-6).

Clarify the specific objectives of the study at the end of the introduction to provide a clearer focus for the reader (lines 10-12).

Response 1:

The authors thank the reviewer for their comments regarding the introduction and have endeavoured to expand this area with further detail on oxygen saturation measurement and inclusion of further relevant studies. We have also expanded on our objective at the end of this section for additional context and clarity.

Comment 2:

Methods

While the methods are generally well-explained, it would be beneficial to include more detail about the statistical analysis, particularly regarding the assumptions underlying the regression analysis and any software used for analysis (lines 14-18).

Please specify how outliers were handled in the dataset, as this can significantly influence the regression results (lines 23-25).

Response 2:

Thankyou for your comments. Please find details of the software used in section “2.4 Statistical Analysis”. We have further expanded this section.

Comment 3:

Results

The results are presented clearly, but the figures could benefit from more comprehensive captions to enhance their interpretability. Ensure that each figure includes all necessary details to understand its significance without referring back to the main text (lines 227-229).

Consider providing a summary table of the key findings from both regression models to facilitate comparison and clarity (lines 238-240).

Response 3:

The authors thank the reviewer for their comments. The figures have been updated to improve interpretability in the context of the wider text and individually. We have also included a table summarising the key findings from both regression models.

Comment 4:

Discussion

The discussion section effectively interprets the findings, but it could be strengthened by addressing the limitations of the study more explicitly. Discuss how these limitations might impact the results and conclusions drawn (lines 268-270).

It would be helpful to explore potential clinical applications of your findings in greater detail. How might this research influence clinical practice, especially regarding non-invasive assessments of coronary sinus oxygen saturation (lines 275-277)?

Response 4:

Thankyou for your comments. The authors have updated the discussion to better clarify the limitations of the study. We have expanded on the discussion regarding clinical applications of the method we have described.

Reviewer 2 Report

Comments and Suggestions for Authors

In the first place, I would like to thank the editor for the opportunity to review this manuscript.

The paper represents a re-evaluation of data from a previously published original scientific work.

The work itself is written very scientifically, is well-readable, and is precisely crafted.

However statistical evaluation itself processes data from a group of only 18 patients.

The authors focused on finding a simpler method for measuring saturation in the coronary sinus, specifically through the calculation of venous blood saturation in the pulmonary trunk. From clinical practice, we know that cardiac decompensation leads to a decrease in blood saturation in the coronary sinus as well as a decrease in blood saturation in the venous system of the systemic circulation, and thus also in the pulmonary trunk.

However, to compare correlations or discrepancies in such closely related output parameters between the systemic (systemic + coronary) and coronary circulation, a larger group of patients is needed with direct assessment of parameters in a separate group of patients.

In light of the above, I must conclude that the presented manuscript carries too high a risk of inaccurate data with an excessive risk of unwarranted conclusions.

I would like to encourage the authors to consider a more thorough handling of the topic of the manuscript.

Author Response

Comment:

In the first place, I would like to thank the editor for the opportunity to review this manuscript.

The paper represents a re-evaluation of data from a previously published original scientific work.

The work itself is written very scientifically, is well-readable, and is precisely crafted.

However statistical evaluation itself processes data from a group of only 18 patients.

The authors focused on finding a simpler method for measuring saturation in the coronary sinus, specifically through the calculation of venous blood saturation in the pulmonary trunk. From clinical practice, we know that cardiac decompensation leads to a decrease in blood saturation in the coronary sinus as well as a decrease in blood saturation in the venous system of the systemic circulation, and thus also in the pulmonary trunk.

However, to compare correlations or discrepancies in such closely related output parameters between the systemic (systemic + coronary) and coronary circulation, a larger group of patients is needed with direct assessment of parameters in a separate group of patients.

In light of the above, I must conclude that the presented manuscript carries too high a risk of inaccurate data with an excessive risk of unwarranted conclusions.

I would like to encourage the authors to consider a more thorough handling of the topic of the manuscript.

Response:

The authors would like to thank the reviewer for their comments regarding the manuscript. Despite a thorough and exhaustive review of the published literature, there were no further articles detailing simultaneous measures of coronary sinus and pulmonary artery or right ventricular saturations. As such, we proceeded with this initial analysis, aware of these limitations, and have described these in the discussion.

Round 2

Reviewer 2 Report

Comments and Suggestions for Authors

The work is presented clearly and factually. The main drawback I find is the small group of patients. Conclusions drawn based on the evaluation of 9 patients from another study may lead to misleading statements.

I leave the decision regarding the publication of the work to the Editor.

Author Response

Comment

The work is presented clearly and factually. The main drawback I find is the small group of patients. Conclusions drawn based on the evaluation of 9 patients from another study may lead to misleading statements.

Response

Thank you for your valuable feedback. We would like to clarify that, in our study, 18 recordings were analyzed to assess correlations. We have acknowledged the limitation of the small recruitment size in our manuscript, and we agree that future research should focus on external validation of these findings to strengthen the conclusions. We appreciate your consideration of these points.